# On the Structural Stochasticity in Graph Neural Networks

## Abstract

Graph neural networks (GNNs) that aggregate and transform point masses as *messages* manifest a wide array of symptoms including limited expressiveness, over-smoothing, and over-squashing. When stochasticity is injected into the structure of the graph, these problems can be jointly remedied, as shown in the unifying framework herein, which theoretically justifies the superior performance of a number of GNN architectures that incorporate random regularization. For the first time, we discover that simple GNNs can *exceed* the power of the Weisfeiler-Lehman test when equipped with structural stochasticity. With insights drawn from the theoretical arguments, we design a principled way to quantify the structural uncertainty in GNNs via variational inference, termed Bayesian Rewiring of Node Networks (BRONX), and showcase its competitive performance with real-world experiments.

## 1 Introduction: Graph Neural Networks (GNNs) and Limitations

Graph neural networks (GNNs)—neurally parametrized models aggregating and transforming node embeddings based on topological neighborhoods—have shown promises in a wide range of domains (Kipf & Welling, 2016; Xu et al., 2018; Gilmer et al., 2017; Hamilton et al., 2017; Battaglia et al., 2018). The key inductive biases of *spatial* (Wu et al., 2019b) GNNs rely on the double rôle of the graph they operate on—it is both an input feature as well as a compute graph. When this graph is rigid, with static (and potentially arbitrarily defined) structure, GNNs display a plethora of problems, including over-fitting (poor generalizability to unseen data), over-smoothing (Oono & Suzuki, 2019; Cai & Wang, 2020; Rusch et al., 2023) (node similarity approaches infinity), over-squashing (information for exponentially-growing receptive field gets over-compressed), under-reaching (Alon & Yahav, 2020) (with finite rounds of message-passing, long-term interactions cannot be realized), and limited expressiveness (Xu et al., 2018) (not able to distinguish between nodes with distinct environments).

A line of fruitful research focuses on injecting stochasticity into the structure of the graph, as a way of regularization aiming at alleviating over-smoothing and over-fitting: Zhang et al. (2018) regards the input graph as a realization of a class of graphs generated by some random processes; Chen et al. (2018) (FastGCN) randomly masks out input graphs under Bernoulli distribution; Rong et al. (2019) (DropEdge) randomly removes edges of input graphs; Hasanzadeh et al. (2020) (Graph DropConnect, GDC) similarly removes edge, although edges are removed independently for each feature dimension; Feng et al. (2020) (GRAND) takes these structural modifications and performs multiple steps of message-passing before aggregating the samples. In addition, the time-tested graph attention networks (GAT) (Veličković et al., 2018), with DropOut (Gal & Ghahramani, 2016) on the attention scores, *de facto* modifies the graph structure randomly. This class of models, referred to as *stochastic* GNNs henceforth, empirically show decent performance with intriguing properties partially remedying the aforementioned problems.

**Main contributions.** In this paper, we hope to further deepen the understanding of *stochastic* GNNs by: (a) providing a theoretical framework justifying how structural uncertainty alleviates over-smoothing, over-squashing, and limited expressiveness; most interestingly, we discover, for the first time, that stochastic GNNs can exceed the discriminative power of the Weisfeiler-Lehman tests. (b) incorporating these insights to design a model—termed Bayesian Rewiring of Node Networks

(BRONX)—to quantify structural uncertainty using variational inference (VI), which, quantifies edge uncertainty *individually* in an amortized manner. To the best of our knowledge, this is the first model capable of quantifying edge uncertainty in transductively (across unseen environments and graphs), since the edges weights are individually learned based on the graph environment.

## 2 PRELIMINARIES: (STOCHASTIC) GRAPH NEURAL NETWORKS

**Graph.** A graph is defined as a tuple of collections of nodes and edges $\mathcal{G} = \{\mathcal{V}, \mathcal{E}\}$. The nodes can be attributed with features $\mathbf{x}_1, \mathbf{x}_2, ..., \mathbf{x}_N = \mathbf{X} \in \mathbb{R}^{N \times C}$, where $N = |\mathcal{V}|$ is the cardinality of the node set and C the feature dimension. An adjacency matrix $A \in \mathbb{R}^{N \times N}$ (usually sparse) associates the edges with nodes:

$$A_{ij} = \begin{cases} 1, (v_i, v_j) \in \mathcal{E}; \\ 0, (v_i, v_j) \notin \mathcal{E}. \end{cases} \tag{1}$$

**Graph neural networks.** A graph neural network can be most generally defined as one adopting a layer-wise updating scheme that aggregates representations from a node's neighborhood and updates its embedding:

$$\mathbf{X}'_v = \phi(\mathbf{X}_v, \rho(\mathbf{X}_u, \hat{A}_{uv}, u \in \mathcal{N}(v))), \tag{2}$$

where $\phi, \rho$ are the *update* and *aggregate* function, repsectively. Omitting the nonlinear transformation step $\phi$ ubiquitous in all neural models, and assuming a convolutional *aggregate* function, $\rho = \text{SUM}$ or $\rho = \text{MEAN}$, a graph neural network layer is characterized by the aggregation/convolution operation that pools representations from neighboring nodes, forming an intermediary representation $\mathbf{X}'$, which on a global level, with activation function $\sigma$ and weights $W$, can be written as:

$$\mathbf{X}' = \sigma(\hat{A}\mathbf{X}W) \tag{3}$$

A GNN *model*, which is a stack of GNN *layers*, can then be presented as

$$\mathbf{X}^{(l)} = \underbrace{\sigma(\hat{A}\sigma(\hat{A}\sigma(\hat{A}...\sigma(\hat{A}}_{l\ \texttt{times}} \mathbf{X} \underbrace{W...W)W)W)}_{l\ \texttt{times}} \tag{4}$$

The primary difference among architectures amounts to distinct effective adjacency matrix $\hat{A}$. Graph convolutional networks (Kipf & Welling, 2016) (GCN) normalizes $\hat{A}$ by the in-degree of nodes $D_{ii} = \sum_j A_{ij}$; graph attention networks (Veličković et al., 2018) (GAT) takes $\hat{A}$ as the attention score; simplifying graph convolution (SGC) (Wu et al., 2019a) taking the normalized adjacency matrix and raise to $K$-th power; and graph neural diffusion (GRAND) takes the matrix exponential of the attention score matrix in GAT (albeit possibly with a different attention form).

$$\hat{A}_{\texttt{GCN}} = D^{-\frac{1}{2}} A D^{-\frac{1}{2}}; \hat{A}_{\texttt{GAT},ij} = \text{Softmax}(\sigma(\text{NN}(\mathbf{x}_i||\mathbf{x}_j))); \hat{A}_{\texttt{SGC}} = \hat{A}_{\texttt{GCN}}^K; \hat{A}_{\texttt{GRAND}} = \exp \hat{A}_{\texttt{GAT}} \tag{5}$$

**Stochastic graph neural networks.** Randomly perturbing the adjacency matrix $A$ (or its normalized / transformed version $\hat{A}$) has been shown to effectively regularize the underlying graph neural network. With the stack of adjacency matrix for all latent features $\mathbf{A} = \bigoplus_{c=0}^{C} A_c \in \mathbb{R}^{C,N,N}$ explicitly written out, and $q$ denoting a (hyper)parametrized Bernoulli distribution, FastGCN (Chen et al., 2018), which masks out nodes (all entries in that column / row) in graph convolution, where DropEdge (Rong et al., 2019) adopts a sparse mask on the last two dimensions:

$$\mathbf{A}_{\texttt{FastGCN}[:,:,v][:,v,:]} \sim q(Z) \in \mathbb{R}^N; \mathbf{A}_{\texttt{DropEdge}[:,u,v]} \sim q(Z) \in \mathbb{R}^{|\mathcal{E}|}. \tag{6}$$

Note that GAT (Veličković et al., 2018), with Dropout (Gal & Ghahramani, 2016) modules operating on the attention scores, have the same effect before the $\text{Softmax}$ operator. Graph DropConnect (GDC) (Hasanzadeh et al., 2020), on the other hand, samples edges of graphs independently for each feature:

$$\mathbf{A}_{\texttt{GDC}[c,u,v]} \sim q(Z) \in \mathbb{R}^{C \times |\mathcal{E}|} \tag{7}$$

Hasanzadeh et al. (2020) also explores VI scheme for GNNs, though different from us, the edge uncertainty is globally learned and is thus not suitable for transductive learning.

**Bayesian framework for graph structural uncertainty.** Transitioning from the algebra of point masses to random variables under certain distributions, we put the dynamics of GNNs into a probabilistic framework. Assuming the (effective) adjacency matrix $\hat{A}$ adopting a (tractable or intractable) distribution $\hat{A} \sim q(\hat{A})$, and the neural network parameter $\theta$ fixed, the probability distribution of the output signal $y$ can be written as

$$p(\mathbf{y}, \hat{A}|\mathbf{X}, \theta) = q(\hat{A})p(\mathbf{y}|\hat{A}, \mathbf{X}, \theta). \tag{8}$$

Marginalizing over the possible structures $\int d\hat{A}$, the predictive distribution becomes

$$p(\mathbf{y}) = \int q(\hat{A})p(\mathbf{y}|\hat{A}, \mathbf{X}, \theta)d\hat{A} = \mathbb{E}_{\hat{A} \sim q(\hat{A})}p(\mathbf{y}|\hat{A}, \mathbf{X}, \theta). \tag{9}$$

For previously surveyed architectures, with the exception of the non-transductive case in Hasanzadeh et al. (2020), $q$ is always chosen as a fixed prior. While training, similar to uncertain weights (Blundell et al., 2015), one uses Markov Chain Monte Carlo (MCMC) to draw unbiased estimates to estimate the parameter gradient $-\partial p(\mathbf{y})/\partial \theta$. During inference, Equation 9 is used (explicitly or implicitly) to compute the expectation of the posterior distribution. Note that, for the family of distributions discussed here, we only consider the multiplicative noise applied on the original sparse adjacency matrix $A$; in other words, different from Zhang et al. (2018), we consider a simpler case where edge strength are perturbed, but no edge is added into the original adjacency matrix.

$$\hat{A} = A \odot Z, Z \sim q(Z) \tag{10}$$

## 3 Theory: Structural stochasticity jointly alleviates unique problems in GNNs—limited expressiveness, over-smoothing, and over-squashing.

It has been extensively studied that GNNs, in addition to the common issues of (over-parametrized) neural networks, display some symptoms unique to the convolutional scheme characterized by Equation 3. In this section, following a survey of these problems, we theoretically show that structural stochasticity alleviates these problems simultaneously.

### 3.1 Structural stochasticity increases expressiveness—the ability to distinguish non-isomorphic graph environments.

First, we reëstablish the notion of *expressiveness* under the probabilistic framework as the ability to render, from distinct inputs, signals that are not *equal in distribution*. This guarantees distinctiveness *in expectation* after nonlinear transformation and can be reflected in the final output of the model:

**Lemma 1.** *There exists some element-wise function $\sigma$ such that*

$$\mathbb{E}_{X \sim q_X}(\sigma(X)) \neq \mathbb{E}_{Y \sim q_Y}(\sigma(Y)) \tag{11}$$

*if $X \sim q_X$ and $Y \sim q_Y$ are not equal in distribution.*

One example of such activation function $\sigma$ is a switch function that is only positive in the region where $X > Y$ and zero everywhere else. With this tool handy, we show that many traditional operations in the algebra of point masses would yield degenerate results actually result in signals not *equal in distribution*. Intuitively, for node representations to achieve equality in distribution is difficult as it requires exactly the same dependence structure within the neighborhood, as we show in the following section.

**Problem: GNNs are at most as expressive as Weisfeiler-Lehman tests.** Xu et al. (2018) has groundbreakingly illustrated that GNNs cannot distinguish graphs (or node neighborhoods) that Weisfeiler-Lehman isomorphism tests (Weisfeiler & Leman) cannot:

**Lemma 2 from Xu et al. (2018).** *Let $\mathcal{G}_1$ and $\mathcal{G}_2$ be any two non-isomorphic graphs. If a graph neural network $\mathcal{G} \rightarrow \mathcal{R}^C$ maps $\mathcal{G}_1$ and $\mathcal{G}_2$ to different embeddings, the Weisfeiler-Lehman graph isomorphism test also decides $\mathcal{G}_1$ and $\mathcal{G}_1$ are not isomorphic.*

**Remedy: With structural stochasticity, GNNs can be more expressive than Weisfeiler-Lehman tests.** The ring-size realization experiment (See Figure 1) is perhaps the poster child for the inability of both classical rank-1 (Morris et al., 2018) GNNs and the Weisfeiler-Lehman test to distinguish non-isomorphic graphs. For two differently-sized rings composed of identical nodes, no matter how many steps of message-passing are conducted, each node is going to aggregate representations from adjacent, identical nodes and cannot yield a different embedding. This is not the case if we, again, switch to the algebra of random variables. Under the probabilistic framework, the graph on which the GNN is operating on, apart from the dual rôle as the computation graph and input feature, is also endowed with a third rôle as the *belief network* in a Bayesian graphical model. After $l$ rounds of message passing, if there exists a ring with a size smaller than $l$, the belief network would show a closed loop, whereas if there are no rings or only rings larger than $l$, the belief network would receive a tree of independent variables, resulting in two node representations not *equal in distribution*. We formalize this thought experiment as:

**Theorem 1** (GNNs with structural stochasticity are more expressive than WL test.). *There exist graphs $\mathcal{G}_1$ and $\mathcal{G}_2$ with adjacency matrix $A_1, A_2$, that are labeled as isomorphic by the Weisfeiler-Lehman test, and some multiplicative distribution $Z \sim q(Z) \in \mathbb{R}^{N \times N}$, such that*

$$\mathbb{E}_{Z \sim q} \text{GNN}(A_1 \odot Z) \neq \mathbb{E}_{Z \sim q} \text{GNN}(A_2 \odot Z) \tag{12}$$

*where* GNN *is a graph neural network architecture with the form in Equation 2.*

*Proof.* One such example is a GNN with MAX aggregate function and update function copying the neighborhood:

$$\mathbf{X}'_v = \text{MAX}(\hat{A}_{uv}, \mathbf{X}_u, \mathbf{X}_v, u \in \mathcal{N}(v)) \tag{13}$$

with $Z \sim q$ some non-zero continuous distribution perturbing the structure with $\hat{A} = a \odot Z$ and initial $\mathbf{X}^{(l=0)} = \mathbf{0}$. If one has two cyclic graphs $\mathcal{G}_1, \mathcal{G}_2$ with ring size $r_1 < r_2 - 1$, which are not distinguishable with the Weisfeiler-Lehman test. At $l = r_1 + 1 < r_2$, we have

$$\mathbb{E}_{Z \sim q} \mathbb{1}(\mathbf{X}_v = \hat{A}_{uv}, u \in \mathcal{N}(v)) > 0 \tag{14}$$

(where $\mathbb{1}$ is the indicator function) holds for $\mathcal{G}_1$ but not $\mathcal{G}_2$. $\qquad\square$

We believe that this proof is concise and interesting enough to be included in the main text here—it exemplifies the difference in distribution resulting from a different dependence graph. In particular, Equation 14 only holds when the sample sample was passed around to the source node. Crucially, this proof, and Theorem 1 would not hold if the structure $A$ were resampled—the parent nodes of all node distribution will be fresh and independent, and belief network depth is reduced to a single layer. We also remark that the argument in Barceló et al. (2020) still holds that GNNs, with or without stochasticity, can only distinguish graph environments at most $K$ edges away when operating $K$ times, and that stochastic GNNs of $K$-layers are at most as powerful as WL tests with $K$ steps. This incentivizes us to design GNNs with more steps, especially since we show that over-smoothing and over-squashing can also be remedied in Section 3.2 and Section 3.3. These principles will the design of our VI-based infinitely-deep framework in Section 4.

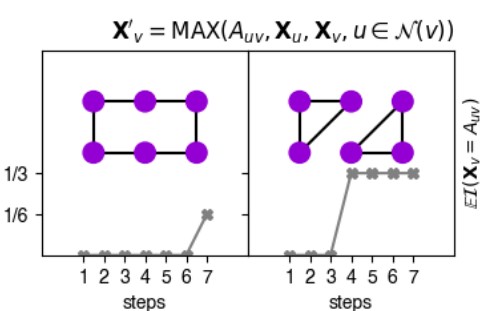

Figure 1: Proof of **Theorem** 1, illustrated.

**Problem: with one aggregator, GNNs are not as expressive as Weisfeiler-Lehman tests.** Xu et al. (2018) also argues that, the key to designing maximally powerful—that is, as powerful as Weisfeiler-Lehman test—GNNs, lies in the choice of aggregation functions ($\rho$ in Equation 2). If $\rho$ is injective, for instance, the SUM function, GNNs can achieve this discriminative power (**Theorem 3** in Xu et al. (2018)). Corso et al. (2020), however, states that such a theorem only holds if the underlying space for the neighborhood multiset (a set with possibly repeating elements) is countable. For the more general continuous space scenario, one aggregator does not suffice:

**Theorem 1 from Corso et al. (2020).** *In order to discriminate between multisets of size $N$ whose underlying set is $\mathbb{R}$, at least $N$ aggregators are needed.*

**Remedy: With structural stochasticity, *one* aggregator suffices to distinguish among multisets with continuous support.** We show that **Theorem 1 from Corso et al. (2020)** will not hold if each element in the neighborhood multiset $\{u \in \mathcal{N}(v)\}$ is multiplied with a stochastic edge weight $Z$ under some distribution (proof in Appendix Section B.1).

**Theorem 2** (One aggregator is sufficient to distinguish multisets.). *Given two arbitrary-sized multisets $X$ and $Y$ with support $\mathbb{R} \setminus 0$, there exists some aggregator $\rho : \mathbb{R}^{|X|} \setminus \mathbf{0} \to \mathbb{R}$ and a scalar distribution $q(Z) \in \mathbb{R}^1$ such that*

$$\rho(zx, x \in X, z \sim q(Z)) = \rho(zy, y \in Y, z \sim q(Z)) \tag{15}$$

*are equal in distribution i.f.f. $X = Y$.*

To compare this with **Theorem 1 from Corso et al. (2020)**, one can regard the rôle of structural stochasticity as packing the expressiveness of multiple aggregators into one.

### 3.2 STRUCTURAL STOCHASTICITY ALLEVIATES OVER-SMOOTHING.

**Problem: Node becomes too similar after rounds of message-passing.** Since message passing, or graph convolutions (Equation 3), behaves like Laplacian smoothing (ignoring the linear and non-linear transformations), after sufficient rounds, the similarities among nodes drastically increase, until the node embedding is only dependent upon their degree. Cai & Wang (2020); Rusch et al. (2023) quantitatively characterizes this scenario via Dirichlet energy among node embeddings $\mathbf{X} \in \mathbb{R}^{N \times C}$:

$$\mathcal{E}(\mathbf{X}) = \frac{1}{N} \sum_{(u,v) \in \mathcal{E}_{\mathcal{G}}} ||\mathbf{X}_u - \mathbf{X_v}||^2, \tag{16}$$

which decreases, sometimes exponentially to approach zero (**Lemma 3.1, 3.2** in Cai & Wang (2020)), with the node representation embedded on a thin subspace of $\mathbb{R}^C$, dependent only upon the node degree not on the initial embedding, as the number of GNN layers increases.

**Remedy: Structural stochasticity delays the decay in Dirichlet energy.** That to introduce random regularization in graph edges or nodes remedies over-smoothing has been empirically studied in Rong et al. (2019); Hasanzadeh et al. (2020). We here argue that such a remedy is universal across nearly all random perturbations on graph structure, regardless of the specific choices of noise structure, and can delay the decay of Dirichlet energy in the convolutional step. Before we continue onto the theoretical setup, we first restrict ourselves to a particular form of GNN as Equation 3, which still entails all architectures surveyed in Section 2. With this thinly narrowed design space, we formalize this argument:

**Theorem 3** (Structural stochasticity delays the decay of Dirichlet energy.). *For a graph $\mathcal{G}$ with adjacency matrix $A \in \mathbb{R}^{N \times N}$ and node embedding $\mathbf{X} \in \mathbb{R}^{N \times C}$, any non-negative, per-element independent noise distribution $Z \sim q \in \mathbb{R}^{N \times N}$ applied on the non-zero entries in the sparse adjacency matrix, $\hat{A} = Z \odot A$, the expectation of Dirichlet energy of the graph convolution operation is greater than or equal to that resulting from the expectation of $\hat{A}$:*

$$\mathbb{E}_{Z \sim q} \mathcal{E}(\hat{A}\mathbf{X}) \geq \mathcal{E}(\mathbb{E}_{Z \sim q}(\hat{A})\mathbf{X}). \tag{17}$$

Theorem 3 states that, within a single graph Laplacian transformation, the decay of Dirichlet energy is slowed by having a stochastic graph structure, which also corresponds to the empirical findings of the stochastic architectures surveyed in Section 2. By induction, this delay also applies to iterative convolution (as shown in Figure 2).

**Corollary 3.1** (Effects of activation function, weights, and multiple rounds.). *The condition in Equation 17 still holds when one or more operations are applied: projection with weights $\mathbf{W} \in \mathbb{R}^{C \times C'}$, convex activation functions, or multiple ($K$) rounds of message-passing:*

$$\mathbb{E}_{Z \sim q} \mathcal{E}(\sigma(\hat{A}^K \mathbf{X}\mathbf{W})) \geq \mathcal{E}\sigma((\mathbb{E}_{Z \sim q}\hat{A})^K \mathbf{X}\mathbf{W}). \tag{18}$$

In plainer words, the graph convolution with structure perturbed by such distribution $q$ is expected to be less smooth and converge to the subspace independent of the initial features of graphs slower.

**Corollary 3.2** (Jensen's gap for Equation 17). *If $Z$, and thereby $\hat{A}$, is per-element independent,*

$$\mathbb{E}_{Z \sim q}\mathcal{E}(\hat{A}\mathbf{X}) - \mathcal{E}(\mathbb{E}_{Z \sim q}(\hat{A})\mathbf{X}) =$$
$$\frac{1}{N} \sum_{(u,v) \in \mathcal{E}_{\mathcal{G}}} \sum_{0 < c \leq C} \sum_{u_{\mathcal{N}} \in \mathcal{N}(u)} \sum_{v_{\mathcal{N}} \in \mathcal{N}(v), v_{\mathcal{N}} \neq u_{\mathcal{N}}} (\text{Var}\hat{A}_{v_{\mathcal{N}}v}\mathbf{X}^2_{v_{\mathcal{N}}} + \text{Var}\hat{A}_{u_{\mathcal{N}}u}\mathbf{X}^2_{u_{\mathcal{N}}}) \quad (19)$$

As such, we can relate the Jensen's gap to the variance of elements of $\hat{A}$— the higher the variance of $\hat{A}$ is, the larger the Jensen's gap, as shown in Figure 2 with LogitNormal distributions with different parameters.

### 3.3 STRUCTURAL STOCHASTICITY ALLEVIATES OVER-SQUASHING.

**Problem: Information loss when representing large neighborhood.** GNNs have *exponentially* growing receptive field as the convolution operations increase, as opposed to the *linear* case for the recurrent neural networks (RNN) (Cho et al., 2014) or convolutional neural networks (CNN) (Krizhevsky et al., 2012), thanks again to the duality of the input / compute graph. Alon & Yahav (2020) observe that, as one stacks multiple GNN layers, the size of the neighborhood and the number of possible combinations of faraway neighbors would soon outgrow the maximum enumerations in a fixed-length floating-point vector. Topping et al. (2022) quantifies such effect by the inter-node Jacobian and relates it to the ($l + 1$ with $l$ being the number of layers) power of the effective adjacency matrix:

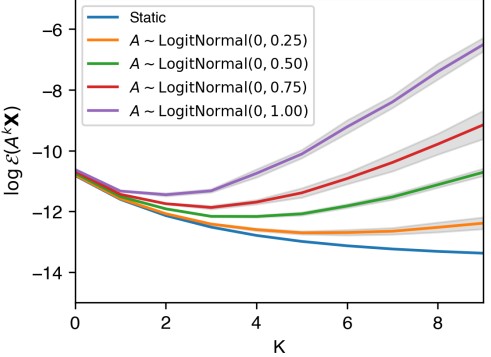

$$|\frac{\partial \mathbf{X}_v^{(l+1)}}{\partial \mathbf{X}_u}| \leq |\nabla\phi|^{(l+1)}(\hat{A}^{l+1})_{uv}. \quad (20)$$

Figure 2: Structural stochasticity delays the decrease of Dirichlet energy on Cora citation graph: Log Dirichlet energy plotted against consecutive graph convolution with adjacency matrix with various edge distributions (without applying weights).

Naturally, if the Jacobian is vanishing, the change in the neighborhood is too small to be reflected in the fixed-lengthed vector and the node is only capable of receiving signals or more immediate neighbors.

**Remedy: Structural stochasticity amplifies squashed signals.** We show that a random perturbation alleviates over-squashing as well; with certain class of activation functions, it slows the decay of inter-node Jacobian magnitude:

**Theorem 4** (Structural uncertainty alleviates inter-node vanishing Jacobian). *For a graph $\mathcal{G}$ with adjacency matrix $A \in \mathbb{R}^{N \times N}$ and node embedding $\mathbf{X} \in \mathbb{R}^{N \times C}$, any noise distribution $Z \sim q \in \mathbb{R}^{N \times N}$ applied on the non-zero entries in the sparse adjacency matrix $\hat{A} = Z \odot A$, the expectation of Jacobian of multiple rounds of message passing is greater than or equal to that resulting from the expectation of $A$:*

$$\mathbb{E}_q|\frac{\partial \hat{A}^K \mathbf{X}}{\partial \mathbf{X}}| \geq |\frac{\partial (\mathbb{E}_q\hat{A})^K \mathbf{X}}{\partial \mathbf{X}}|. \quad (21)$$

The proof (See Appendix Section B.3), again, relies on Jensen's inequality. We can intuitively explain this result as follows: with random perturbation, the representation capacity of a fixed-length vector increases. During convolution, when two neighbor distributions are combined, the resulting distribution has a higher information content, which, if measured by Shannon entropy, does not dissolve:

$$H(\mathbf{X}_v) = \sum_{u \in \mathcal{N}(v)} (H(\hat{A}_{uv}) + H(\mathbf{X}_u)) \quad (22)$$

if the neighboring edge representations are independent. This is in contrast with the fixed-length vector, which has a fixed number of possible combinations, or fixed entropy.

**Corollary 4.1.** *(Effects of activation function and weights) The condition in Equation 21 still hows with weights* $\mathbf{W} \in \mathbb{R}^{C \times C'}$; *additionally, this still holds after activation function* $\sigma$ *with convex first-order derivatives*

$$\mathbb{E}_q | \frac{\partial \hat{A}^K \mathbf{X} \mathbf{W}}{\partial \mathbf{X}} | \geq | \frac{\partial (\mathbb{E}_q \hat{A})^K \mathbf{X} \mathbf{W}}{\partial \mathbf{X}} |; \mathbb{E}_q | \frac{\partial \sigma(\hat{A}^K \mathbf{X} \mathbf{W})}{\partial \mathbf{X}} | \geq | \frac{\partial \sigma((\mathbb{E}_q \hat{A})^K \mathbf{X} \mathbf{W})}{\partial \mathbf{X}} |. \tag{23}$$

To sum up this section, through **Theorems 1 ∼ 4**, we have insofar characterized the promising theoretical properties of structural stochasticity, which have sporadically appeared in previous literature (Hasanzadeh et al., 2020; Rong et al., 2019) used on various models but have never been studied systematically to justify the efficacy of stochastic GNNs. The most useful mathematical tools in this section are the stringent condition of equality in distribution and Jensen's inequality.

Finally, it is worth noting that controlled stochasticity as a regularization also addresses overfitting. However, one can verify that none of the herein studied theorems would hold if merely *parameter uncertainties*, as in weight perturbation or activation dropout, were applied. We also experimentally test this in Section 5. This suggests that *structural uncertainty* has a unique efficacy on the dynamics of graph neural networks.

## 4 ARCHITECTURE: BAYESIAN REWIRING OF NODE NETWORKS (BRONX)

**Design principles.** From the theoretical analysis in Section 3, we extract several qualitative insights for the design of maximally powerful stochastic graph neural networks: (a) One aggregator, for example SUM, is sufficient for stochastic GNNs, as long as followed by non-linear activation functions (**Theorem** 2); (b) More message-passing rounds corresponds to higher expressive power over larger receptive fields (**Theorem** 1), without leading to dramatic over-squashing and over-smoothing, though the reuse of edge samples must be guaranteed; (c) Edge diversity and anisotropy remedies over-smoothing (**Theorem** 3); (d) Inter-layer activation functions should have convex first derivatives, (**Theorem** 4).

With these principles, we design an infinitely-deep graph neural networks with edge uncertainty and anisotropy controlled through a variational inference framework. Since the infinitely-deep GNN resembles the graph diffusion or a graph rewiring process (Klicpera et al., 2019), we term our model Bayesian Rewiring Of Node networKS, or BRONX for short.

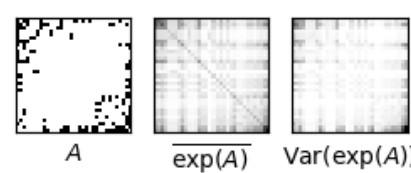

Figure 3: Illustration of stochastically rewired structure with Karate club graph (Zachary, 2002).

**Infinitely-deep graph neural networks.** From our theoretical analysis, we find that, with structural stochasticity alleviating over-squashing and over-smoothing, depth only correlates with larger reception field and higher expressiveness. We therefore apply Equation 3 infinitely many times with residual connection and without channel mixing, arriving at a continuous variant of a graph neural networks (Xhonneux et al., 2019; Chamberlain et al., 2021)

$$\frac{\partial \mathbf{X}(t)}{\partial t} = \hat{A}(t) - I. \tag{24}$$

which simulates a diffusion process on the graph manifold. If $\hat{A}$ remains constant during the course of the diffusion, to reduce variance and increase expressiveness (**Theorem** 1), Equation 24 has an analytical solution:

$$\mathbf{X}(t) = \exp(\hat{A} - I)\mathbf{X}(t = 0). \tag{25}$$

Since the matrix exponential can also be written as $\mathbf{X}(t) = \sum_{k=0}^{+\infty} \frac{1}{k!} \mathbf{X}^k(t = 0)$, as the sum of an infinite series of $k$-round message passing, the conditions discussed in Section 3 still holds. This process can also be seen as a way of rewiring to produce samples of rewired subgraphs, as illustrated in Figure 3.

**Edge distribution class.** It has been studied that the sparsity of effective graph adjacent matrix would further regulate message-passing. Bernoulli distribution, however, are difficult to parametrize in a differentiable, amortized manner without introducing additional variances or biases (Hasanzadeh et al., 2020). We here propose an alternative class of edge distribution—Logit Normal distribution that are fully parametrizable and differentiable using the resampling trick (Kingma et al., 2015):

$$Z \sim \mathcal{N}(\mu, \Sigma); A = \sigma(q) = \frac{1}{1 + \exp(-Z)} \sim \text{LogitNormal}(\mu, \Sigma), \tag{26}$$

which resembles a Bernoulli distribution when $\Sigma >> 0$, i.e. when the edge uncertainty is high.

**Variational inference framework.** Now it only remains to provide a variational parametrization to the (time-independent) edge distribution and combine it with with the initinitely deep graph neural networks to complete the model (Equation 9). From the initial node distribution $\mathbf{X}(0)$, we parametrize the logit-normal distribution (Equation 26) using dot-product attention with key weight shared $\phi = \{\mu, \Sigma\}$:

$$\mu_{uv} = (W_k X_u)^T W_\mu \mathbf{X}_v; \Sigma_{uv} = (W_k \mathbf{X}_u)^T W_\Sigma \mathbf{X}_v. \tag{27}$$

We furthermore purpose a prior for the edge weights conditioned on the source nodes:

$$A_{uv}|\mathbf{X}_u \sim q(A_{uv}|\mathbf{X}_u) = \mathcal{N}(\mu_u, \Sigma_u), \tag{28}$$

with the parameters given by a linear transformation of the node embedding. As such, with training data pair $(\mathbf{X}, y)$, we can maximize a data evidence through the *evidence lower bound* (ELBO) given as (ignoring neural network parameters for now)

$$\mathcal{L}(\phi) = \mathbb{E}_{Z \sim q(A|\mathbf{X};\phi)}[\log(y|A, \mathbf{X}) - q(A|\mathbf{X}; \phi)], \tag{29}$$

with the likelihood term corresponding to the final task, namely regression or classification. A general recipe to contruct losses given this is to decent $-\mathcal{L}(\phi)$.

**Comparison with other stochastic GNNs.** The BRONX architecture is distinct from other stochastic GNNs in several ways: (a) To promote edge anisotropy, the edge distribution is learned independently for each edge (Insight (b) from **Design principles**). (b) With the over-squashing and over-smoothing behaviors remedied, and expressiveness scale with the number of message-passing rounds, we employ a continuous, infinitely deep graph neural network (Insight (c) from **Design principles**).

**Complexity.** The space and runtime complexity is controlled by evaluating the graph diffusion step Equation 24, $\mathcal{O}(|\mathcal{E}|k)$, where $|\mathcal{E}|$ the number of edges and $k$ the number of samples.

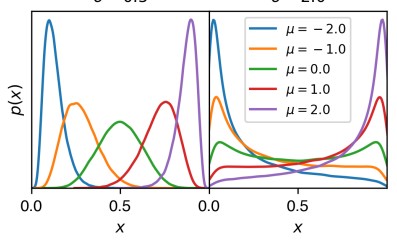

## 5 EXPERIMENTS

We benchmark the performance of the BRONX model designed in Section 4 to illustrate its competitive performance, and more importantly, understand how various design principles outlined in the theoretical framework (Section 3) contribute to the performance.

Figure 4: An illustration of the shape of LogitNormal distribution.

**Dataset.** We benchmark our model on the popular Planetoid citation datasets (Yang et al., 2016), as well as the coäuthor (Shchur et al., 2018) and copurchase (McAuley et al., 2015) datasets common in social modeling. While acknowledging its limitations (lack of long-range interactions, small in size, some linear relationship) (Dwivedi et al., 2020), we argue that this small dataset is the only common task every popular model has been tested against. At the same time, the original train–validation–test split has also been adopted.

|  | Cora | CiteCeer | PubMed | Coauthor CS | Photo |
|---|---|---|---|---|---|
| GCN (Kipf & Welling, 2016) | 81.5 | 70.3 | 79.0 | 91.1 ± 0.5 | 01.2 ± 1.2 |
| GAT (Veličković et al., 2018) | 83.0 ± 0.7 | 72.5 ± 0.7 | 79.0 ± 0.3 | 91.3 ± 0.1 | 90.9 ± 0.3 |
| GRAND (Chamberlain et al., 2021) | 84.7 ± 0.6 | 73.6 ± 0.3 | 81.0 ± 0.4 | 92.9 ± 0.4 | 92.4 ± 0.8 |
| DropEdge (Rong et al., 2019) | 82.8 | 72.3 | 79.6 |  |  |
| GDC (Hasanzadeh et al., 2020) | 82.20 | 71.72 |  |  |  |
| GRAND (Feng et al., 2020) | 85.4 ± 0.4 | 75.4 ± 0.4 | 82.7 ± 0.6 |  |  |
| BRONX | 85.5 ± 0.3 | 74.3 ± 0.6 | 83.0 ± 0.5 | 94.5 ± 1.0 | 92.8 ± 0.5 |

Table 1: Test accuracy (%) on Cora dataset.

**Results** In sum, BRONX achieves competitive performance when compared with the state-of-the-art graph neural networks surveyed here. Among the baselines, the most performant GRAND (graph random neural networks) (Feng et al., 2020) also celebrates stochasticity as the central innovation in the architecture, though hand-crafted intricate sampling and regularization scheme is employed with samples from the prior alone. We argue that the comparison between BRONX and GRAND (graph neural diffusion) (Chamberlain et al., 2021) (which resembles the maximum-likelihood variant of BRONX) highlights the utility of edge uncertainty quantification. The comparison between DropEdge (Rong et al., 2019; Hasanzadeh et al., 2020) and BRONX underscores the importance of a principledly constructed amortized scheme for edge uncertainty.

**Ablation study.** To cleanly separate the contribution of each innovation proposed in Section 4, we conduct an ablation study benchmark the test accuracy on the Cora dataset. *Maximum likelihood estimate(83.5%)* resembles the dynamics of Chamberlain et al. (2021), which is still a powerful model, though without the benefits of stochasticity. *Structural uncertainty from prior(80.1%)* rewires the graph adjacency matrix simply as its matrix exponential $\exp A$ with no learned component, which justifies its underwhelming performance. *Without conditional prior(83.0%)* switches the conditional prior in Equation 28 with a plain $\mathrm{LogitNormal}(\mathbf{0}, \mathbf{1})$; this would lead to over-regularized edges that does not absorb information from the source. *Weight uncertainty quantification(81.5%)* highlights the difference between weight uncertainty (Blundell et al., 2015) and structural uncertainty—the theoretical benefits presented in Section 3 does not apply to weight uncertainty alone.

## 6 CONCLUSIONS

Random regularization is a ubiquitous trick applied in all schools of neural modeling. The pioneer researchers in graph neural networks have been using random perturbations on various aspects of GNNs with success, although its fundamental rôle might have never been put into a holistic, systemic perspective. The first half of of paper aims to address precisely this and to deepen our understanding of stochastic graph neural networks as a general instrument. Next, we extract insights from the theoretical analysis to guide the practical design of stochastic GNNs and cleanly dissect the utility of each modification.

**Limitations.** *Theoretical.* Relying heavily on the notion of *equality in distribution* and Jensen's inequality, we illustrate some advantages of stochastic GNNs in Section 3 in relation to the common pathologies of GNNs. We nevertheless have not quantitatively characterize the magnitude of these differences, namely the Jensen gap. The condition for **Theorem** 1, for example, might be difficult to satisfy and such signal might loose during numerical integration and optimization. *Architectural.* The focus of this paper is to elucidate the conceptual advantages of stochasticity in GNNs, not to find the best architecture for stochasticity GNNs. As such, we have only benchmarked the simplest model with infinitive depth where the diffusion. Again, the dataset used in this benchmark is, though popular and common, minimalistic.

**Future directions.** We plan to address the remaining open questions in the theoretical framework, to demonstrate a clearer translation between theoretical property and practical utility in a wider range of benchmark tasks. In particular, this uncertainty-aware framework is suitable to regime where data is scarce and model uncertainty can be used to guide the experimental design, for example in drug

discovery. At the same time, we plan to study the rôle of stochasticity more closely under our framework in an equivariant setting (Godwin et al., 2021).

**Social impact.** In this paper we demonstrate a powerful learning scheme over graphs incorporating stochasticity. We hope that this will further the endeavor of us, as a scientific community, to better understand the nature and dynamics of *relations*, be it social or physical. On the other hand, as is with all powerful graph neural networks or machine learning methodologies in general, harmful impact might occur if it were used in illegal and immoral contexts, for instance the aggressive feed catering in social network platforms, or the design of overly addictive narcotics.

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

## A  EXPERIMENTAL DETAILS

The model, as well as all evaluation scripts, are distributed open-source in the anonymized repo https://anonymous.4open.science/r/bronx-5F7F/README.md. It is implemented in PyTorch (Paszke et al., 2019) (tensor-accelerating framework) and Pyro (Bingham et al., 2018) (probabilistic inference). Dormand-Prince method with order 5 is employed to evaluate the integral in Equation 24 since the analytical solution (Equation 25) is not tractable. Like in Chamberlain et al. (2021), the attention parameters determining the edge distributions are initialized as small constants, to ensure diffusion stability. Hyperparmeters, including the network size, the optimization schedule, the diffusion time, is tuned using Ray Tune (Liaw et al., 2018).

## B  PROOFS

### B.1  PROOF OF THEOREM 2

We prove Theorem 2 for SUM aggregator on $\mathbb{R} \setminus \{0\}$, although is easy to expand to MEAN and MAX aggregators on $\mathbb{R}^C \setminus \{\mathbf{0}\}$.

*Proof.* Suppose we have two multisets $\mathbf{X} = \{x_i, i = 1, 2, ..., N_X\}$ and $\mathbf{Y} = \{y_j, i = 1, 2, ..., N_Y\}$. We choose the multiplicative noise $Z \sim q = \text{Uniform}(0, 1)$, with the moment generating function $M_Z(t) = \frac{e^t - 1}{t}$, and $\text{SUM}(\xi_q(X))$ and $\text{SUM}(\xi_q(Y))$ are equal in distribution. Thus, the moment generating function of $\text{SUM}(\xi_q(X))$ is

$$M_{\text{SUM}(\xi_q(\mathbf{X}))}(t) = M_{\sum x_i z_i}(t) = \frac{\prod(e^{x_i t} - 1)}{\prod x_i t} \tag{30}$$

Since $\text{SUM}(\xi_q(X))$ and $\text{SUM}(\xi_q(Y))$ are equal in distribution, $M_{\text{SUM}(\xi_q(X))}(t) = M_{\text{SUM}(\xi_q(Y))}(t)$, and therefore $\frac{\prod(e^{x_i t} - 1)}{\prod x_i t} = \frac{\prod(e^{y_j t} - 1)}{\prod y_j t}$. Considering the Taylor expansion of $\exp(\cdot)$, we have

$$\sum x_i^n = \sum y_j^n \tag{31}$$

for any $n \in \mathbb{N}^+$. Since $\forall x_i \neq 0$ and $\forall y_i \neq 0$, we conclude that $\mathbf{X}$ and $\mathbf{Y}$ are equal. $\qquad \square$

## B.2 PROOF OF THEOREM 3

*Proof.*

$\mathbb{E}_{Z\sim q}\mathcal{E}(\hat{A}\mathbf{X})$

$$= \mathbb{E}_{Z\sim q}\frac{1}{N}\sum_{(u,v)\in\mathcal{E}_{\mathcal{G}}}||\hat{\mathbf{A}}\mathbf{X}_u - \hat{\mathbf{A}}\mathbf{X}_v||^2$$

$$= \mathbb{E}_{Z\sim q}\frac{1}{N}\sum_{(u,v)\in\mathcal{E}_{\mathcal{G}}}||\sum_{u_{\mathcal{N}}\in\mathcal{N}(u)}\hat{A}_{u_{\mathcal{N}}u}\odot\mathbf{X}_{u\mathcal{N}} - \sum_{v_{\mathcal{N}}\in\mathcal{N}(v)}\hat{A}_{v_{\mathcal{N}}v}\odot\mathbf{X}_{v\mathcal{N}}||^2$$

$$\geq \frac{1}{N}\sum_{(u,v)\in\mathcal{E}_{\mathcal{G}}}||\sum_{u_{\mathcal{N}}\in\mathcal{N}(u)}\mathbb{E}_{\hat{A}\sim q}(\hat{A})_{u_{\mathcal{N}}u}\odot\mathbf{X}_{u\mathcal{N}} - \sum_{v_{\mathcal{N}}\in\mathcal{N}(v)}\mathbb{E}_{\hat{A}\sim q}(\hat{A})_{v_{\mathcal{N}}v}\odot\mathbf{X}_{v\mathcal{N}}||^2$$

$$\tag{32}$$

$\square$

## B.3 PROOF OF THEOREM 4

$$|\frac{\partial A^K\mathbf{X}}{\partial\mathbf{X}}|_{uv} = (A^K)_{uv} = \sum_{\omega_{uv}\in\Omega_{uv}}\prod_{i\in\omega}A_i; \tag{33}$$

where $\Omega_{uv}$ denotes the collection of walks between $u$ and $v$. If $\omega$ is a path,

$$\mathbb{E}(\prod_{i\in\omega}A_i) = \prod_{i\in\omega}\mathbb{E}(\hat{A})_i, \tag{34}$$

otherwise

$$\mathbb{E}(\prod_{i\in\omega}A_i) \geq \prod_{i\in\omega}\mathbb{E}(\hat{A})_i. \tag{35}$$

