# OpenReview forum: "On the Stochasticity in Graph Neural Networks"
_ICLR.cc/2024/Conference — Submitted to ICLR 2024_

### Official Review · Reviewer_7ZWa · 2023-11-01

**Soundness:** 2 fair
**Presentation:** 2 fair
**Contribution:** 1 poor
**Rating:** 3
**Confidence:** 3

**Summary:**

The manuscript studies the role of structural stochasticity (where edge strength can be perturbed, but no edge is added in the original adjacency matrix) in Graph neural networks (GNNs). Specifically, the authors claim that the limited expressiveness, oversmoothing, and over-squashing problems can be remedied with the adoption of the structural stochasticity. Based upon this observation, the author suggests a design principle of the structure of the GNNs with the stochasticity.

**Strengths:**

The problem is of sufficient interest since limited expressiveness, oversmoothing, and over-squashing problems on GNNs are significant issues that need to be resolved, and empirically, engineers have tried to put randomness to resolve these issues. Hence, the objective is of importance to analyze the role of the structural stochasticity and quantify how much this helps in improving performance.

**Weaknesses:**

1. There are rooms for improvement on the statements of Theorems 3 & 4: The statements do not directly tell us that oversmoothing and oversquashing can be alleviated. Both contain inequalities between the one with stochasticity and the original one. If the difference is marginal, as the number of layers $K$ increases, this would not affect the improvement. For example, if the values of the two are $1/2$ and $1/3$, then $(1/2)^K \approxeq (1/3)^K$ even for the moderate $K$.
2. Technical contributions seem to be rather limited: The proof that the authors relied on is the convexity of the activation functions (or that of the first derivative of the activations), which can be limited in practice. For example, sigmoid and tangent hyperbolic functions are not convex.
3. The stochasticity assumption on the case where edge strength can be perturbed, but no edge is added in the original adjacency matrix, is bit limited.
4. It would have been nice if the assumptions that the authors made is presented in a clearer manner (e.g., by explicitly put "Assumption" separately from the normal texts or Theorem/Lemma statement).
5. Minor typo: p-5 stps -> steps

**Questions:**

1. p-2 $A[:,v, v] \sim q(Z)$: Does this mean that for each feature (i.e., for each column) $A[v, v]$ only becomes zero? or the entire row $A[v, :]$ or column $A[: v]$ become zero?
2. p-5: Does $K$ indicate the number of layers?
3. p-5 in the proof of Theorem 2: For $\ell = r+1$, is $r \in \{r_1, r_2\}$?
4. p-5 in the proof of Theorem 2: Does $\mathcal{I}(\cdot)$ indicate the indicator function?

---

> ### Author Response · Authors · 2023-11-17
> **Jensen's Gap; No convexity assumption; Improved assumption presentation.**
>
> Thank you so much, Reviewer `7ZWa`, for your constructive feedback! In the first stage of our manuscript revision, focusing on the theoretical presentation, we have incorporated your feedback to significantly overhaul the theoretical framework, which, we hope, has addressed your concerns on the novelty of our theoretical contribution.
>
> >The statements do not directly tell us that oversmoothing and oversquashing can be alleviated. Both contain inequalities between the one with stochasticity and the original one. If the difference is marginal, as the number of layers  increases, this would not affect the improvement.
>
> To address your concern, we have derived the Jensen's gap analytically for Theorem 3, which is related to the variance of the edge distribution---the higher variance correspond to slower decay in Dirichlet energy, which we also illustrate in Figure 2. With a sufficient structural noise, the difference is considerable and amplified as the number of layer increases. On the other hand, the Jensen's gap for Theorem 4 can also be related to the moments of the edge distribution, albeit the derivation is less intuitive.
>
> >Technical contributions seem to be rather limited: The proof that the authors relied on is the convexity of the activation functions
>
> We have rewritten Theorem 3 and 4 to cleanly separate the roles of activation functions---they are not necessary conditions, as no activation or identity activation also suffices. On the other hand, concave activation functions like `sigmoid` and `tanh` might indeed break these theorems, so we included "convex activation functions" as one of the design principles in Section 4 (Architecture).
>
> >The stochasticity assumption on the case where edge strength can be perturbed, but no edge is added in the original adjacency matrix, is bit limited
>
> As stated in the Limitation section, the scope of this paper only entails perturbation of edge weights, with no added edge, as that would require more assumptions on the nature of the graph which is not always available, as Zhang et al. 2018 has shown.
>
> >It would have been nice if the assumptions that the authors made is presented in a clearer manner
>
> We have reworked the presentation of the theorems so that the assumptions are more clearly stated at the beginning of the theorems.
>
> >1. p-2 : $A[:, u, v]$ Does this mean that for each feature (i.e., for each column)  $A[v,v]$ only becomes zero? or the entire row  $A[v,:]$ or column $A[:, v]$ become zero?
>
> The later. We have rewritten the confusing paragraph.
>
> >2. p-5: Does  $K$ indicate the number of layers?
>
> Yes! We have clarified.
>
> > 3. p-5 in the proof of Theorem 2: For $l=r+1$, is $r\in r_1, r_2$?
>
> Yes! We have now clarified.
>
> > 4. p-5 in the proof of Theorem 2: Does  indicate the indicator function?
>
> Yes! We have made this clearer in the text now.
>
> Reference:
> YingxueZhang,SoumyasundarPal,MarkCoates,andDenizU ̈stebay.Bayesiangraphconvolutional neural networks for semi-supervised classification, 2018.

---

> ### Comment · Reviewer_7ZWa · 2023-11-22
> **Thank you very much for the detailed rebuttal**
>
> Thank you very much for the detailed rebuttal. I have read the authors' rebuttals and the other reviews. Although the reviewer thinks that the implications of the results are interesting, it would be nice to see more concrete quantifications of Jensen's gap and how the proposed design of model will impact on the performance and addressed issues in a more concrete manner.
> Therefore, I will maintain the score.

---

> > ### Author Response · Authors · 2023-11-23
> > **A note on Jensen's Gap**
> >
> > Thanks, Reviewer `7ZWa`, for your response!
> >
> > Following your suggestion, we have included Corollary 3.2, where we directly linked Jensen's gap for Eq. 17 to the variances of the elements in the effective adjacency matrix.
> > Please let us know how we can further concretely quantify this measure.
> > We would appreciate your further insights.

---

### Official Review · Reviewer_cENw · 2023-11-02

**Soundness:** 3 good
**Presentation:** 2 fair
**Contribution:** 2 fair
**Rating:** 5
**Confidence:** 4

**Summary:**

In this paper, the authors study how using stochasticity can improve GNNs. The model used in the paper is called Bayesian Rewiring of Node Networks (BRONX) which allows to quantify structural uncertainty using variational inference. In Theorem 2, they show how stochasticity can allow to go beyond WL test, thus improving GNNs power. In Theorem 3, how the randomness prevents oversmoothing, and Theorem 4 shows how it prevent oversquashing. The paper is concluded with experiments.

**Strengths:**

- motivated problem
- various experiments

**Weaknesses:**

- the paper is 'slow,' the contributions start from page 4
- the theoretical result while being nice are not enough to address GNN issues

**Questions:**

I believe this is a nice paper but unfortunately the theoretical contributions of the paper are not enough for this venue.


--------------------------------------------
After the rebuttal: I appreciate the authors for their response and revision, as they made a lot of changes to improve the quality of the paper. As they partially addressed my questions/comments, I decided to slightly increase my score.

---

> ### Author Response · Authors · 2023-11-17
> **Overhauled flow; enriched theoretical framework.**
>
> Many thanks again, Reviewer `cENw`, for your constructive feedback. According to your suggestion we have significantly improved the theoretical framework of our paper, as well as its presentation.
>
> >- the paper is 'slow,' the contributions start from page 4
>
> We have overhauled the structure of the paper, with compressed introduction and preliminary sections and a new theoretical presentation framework, where we interweave the problems statements with theoretical evidence. Please let us know if you believe that this has improved the flow and readability of the paper.
>
> >- the theoretical result while being nice are not enough to address GNN issues
>
> We have enriched our theoretical framework with new insights, including the derivation of Jensen's gap for Theorem 3 and the separation of convexity requirement for activation functions. We believe that these new insights justifies the empirical performance of stochasticity-incorporating GNNs and provides new guidelines for stochastic GNN design. We would be grateful if you would elaborate on why you think the previous results are not enough so we can further improve the manuscript to address your concerns.

---

### Official Review · Reviewer_NERp · 2023-11-02

**Soundness:** 3 good
**Presentation:** 3 good
**Contribution:** 3 good
**Rating:** 6
**Confidence:** 2

**Summary:**

The paper focuses on the structural stochasticity in graph neural networks (GNNs) and its benefits in addressing issues like limited expressiveness, oversmoothing, and over-squashing. It provides a theoretical framework justifying how structural uncertainty alleviates over-smoothing, over-squashing, and limited expressiveness in GNNs. It also discovers that stochastic GNNs can exceed the discriminative power of the Weisfeiler-Lehman tests. It introduces a method called BRONX, which quantifies the structural uncertainty in GNNs through variational inference. BRONX showcases competitive performance with other stochastic GNN models like GRAND and DropEdge with real-world experiments, highlighting the importance of edge uncertainty quantification and a principledly constructed amortized scheme for edge uncertainty.

**Strengths:**

1. Originality: The paper introduces the concept of structural stochasticity in graph neural networks (GNNs) and demonstrates its benefits in addressing limitations of traditional GNNs. It also presents the BRONX framework, which quantifies structural uncertainty in GNNs through variational inference, providing a novel approach to address these limitations.

2. Significance: The paper's findings have significant implications for the field of GNNs. By injecting stochasticity into GNN structures, the paper shows that simple GNNs can outperform the Weisfeiler-Lehman test. The introduction of BRONX as a method to quantify structural uncertainty in GNNs also has practical implications for improving GNN performance.

3. Clarity: The paper presents its findings and contributions in a clear and concise manner. It explains the theoretical arguments, the design of BRONX, and the real-world experiments in a way that is easy to understand.

**Weaknesses:**

1. The paper does not thoroughly discuss the potential limitations or drawbacks of the BRONX framework. It would be valuable to address any potential challenges or trade-offs that researchers might encounter when using BRONX in different contexts.

2. The paper lacks a comprehensive comparison with existing stochastic GNN models, limiting the understanding of how BRONX performs in relation to other approaches.

3. The paper does not provide a comprehensive comparison of the proposed method, BRONX, with existing methods for quantifying structural uncertainty in GNNs. Without such a comparison, it is difficult to assess the effectiveness and superiority of BRONX in practical scenarios.

4. This paper only considers a simpler case where edge strength are perturbed, but not cover the increase and decrease of the number of the edges.

**Questions:**

1. Could you provide a more comprehensive comparison of BRONX with existing stochastic GNN models to better understand its performance and advantages?

2. Could you explain what "$\mathcal {I}$" means in equation (20)?

3. Could you discuss any potential limitations or drawbacks of the BRONX framework? Addressing these challenges would enhance the understanding and applicability of the approach.

4. It would be beneficial if you could discuss the scalability of the BRONX framework and its performance on larger datasets. This would provide a better understanding of its practical utility.

5. It would be helpful if you could provide more detailed explanations and examples of the practical implementation of the BRONX framework to assist readers in applying it in real-world scenarios.

6. There may be some minor errors in the Equation (29), which can be corrected.

---

> ### Author Response · Authors · 2023-11-22
> **Emphasized theoretical and architectural limitations; Added Comparison and Complexity sections;**
>
> Thank you, Reviewer `NERp`, for your thorough and detailed review.
> We have incorporated your advice and significantly improved the manuscript.
>
> >The paper does not thoroughly discuss the potential limitations or drawbacks of the BRONX framework.
>
> We have fleshed out the **Limitation** section, discussing the *theoretical* and *architectural* limitations of our framework.
> In particularly, we have reiterated that we are only concerned with edge weight perturbation in this paper, with no new edges being added.
>
> >The paper lacks a comprehensive comparison with existing stochastic GNN models
>
> >The paper does not provide a comprehensive comparison of the proposed method, BRONX, with existing methods for quantifying structural uncertainty in GNNs.
>
> Thanks for pointing this out. We have now emphasized in the paper that, BRONX treats the edges *individually* and assigns different distributions to different edges, which, to the best of our knowledge, is entirely novel and allows transductive learning.
> All previous methods samples the prior distribution for all edges, making transductive learning impossible.
>
> >1. Could you explain what "$\mathcal{I}$" means in equation (20)?
>
> It is the indicator function. We have now clarified this. Thanks again for raising this.
>
> > 2. Could you discuss any potential limitations or drawbacks of the BRONX framework? Addressing these challenges would enhance the understanding and applicability of the approach.
>
> We have reworked the limitations sections to reflect the limitations and future directions.
>
> >3. It would be beneficial if you could discuss the scalability of the BRONX framework and its performance on larger datasets. This would provide a better understanding of its practical utility.
>
> We have now included a complexity analysis---the runtime and storage complexity scales linearly with regard to the number of edges, making it feasible for large systems.
>
> >4. It would be helpful if you could provide more detailed explanations and examples of the practical implementation of the BRONX framework to assist readers in applying it in real-world scenarios.
>
> We have now expanded the Details section in the appendix. The python package is also distributed open-source.
>
> >5. There may be some minor errors in the Equation (29), which can be corrected.
>
> Thank you! We have now corrected the errors!

---

### Author Response · Authors · 2023-11-10
**Thank you for your feedback. We plan to improve the paper in three stages.**

Thank you, Reviewers, for your detailed and constructive feedback. Based on this feedback, we plan to substantially improve the paper from _three perspectives_, which will be reflected in the comments and in the manuscript in stages:

- Presentation.
    - **The tempo of the flow.** Reviewer `cENw` correctly points out that the pace of the paper is rather slow, with main contributions not appearing in the first three pages. We plan to make the introduction sections more concise and overhaul the structure of the paper.
    - **Unclear assumptions.** Reviewer `7ZWa` rightly criticizes that the assumptions of the theorems are buried in the main text, as opposed to clearly highlighted. We plan to sufficiently emphasize these assumptions in our next draft.
    - **Scope.** We plan to make it sufficiently clear in the manuscript that adding new edges are beyond the scope of the paper, which is a limitation of the scope of this paper. We also plan to add a discussion on possible future directions to extend this framework to cases where we add edges to the original graph, to address Reviewers `NERp` and `7ZWa`'s concerns.
- Theoretical.
    - **Convexity assumption.** As `7ZWa` points out, some of our proofs rely heavily on the convexity of activation functions. Although most of the activation functions in the `ReLU` family, which are the most popular class of activation functions, are already convex at least on regions, we use this convexity requirement to guide the design of our model, BRONX. We plan to add more discussion on this point, specifically in terms of the practical implications of this condition.
    - **Jensen's gap.** `7ZWa` finds the qualitative result underwhelming---the alleviation in over-smoothing and over-squashing is presented as inequalities. We propose the following to improve our theoretical results presentation:
	    - We plan to quantify Jensen's gap for these inequalities to quantitatively relate these properties to the strength of noise perturbations of edge weights.
	    - We will experimentally investigate the relationship between the model performance and edge noise distribution.
- Experimental. Incorporating `NERp`'s suggestions, we plan to provide a more comprehensive experimental evaluation of the BRONX framework, especially on real-world modeling tasks.

Please let us know, dear Reviewers, if you believe that these steps would lead to better clarity and readability of the paper, addressing your concerns. We would be grateful to further suggestions!

---

### Author Response · Authors · 2023-11-23
**Thank you, all reviewers!**

Many thanks again, all reviewers, for your detailed and thorough feedback. During the passing ten days, we have incorporated your suggestions to significantly improve the manuscript, hoping that it has now addressed your concerns.

## Main contribution recap
First, let us recap the main contribution of the paper:
- We have proposed a general framework characterizing the impact of structural stochasticity in graph neural networks, which justifies the empirical success of a plethora of GNN models that employ random regularization.
- Under this framework, we have theoretically shown how structural uncertainty jointly alleviates the pressing issues in GNN design---limited expressiveness, over-smoothing, and over-squashing.
- Harnessing the insights from the theoretical framework (with over-smoothing and over-squashing alleviated, deeper stochastic GNNs only correspond to more expressive models), we design BRONX, an infinitely deep graph diffusion model, and showcase its utility in real-world datasets.

Most notably, for the first time, we show that simple GNNs can exceed the expressive power of Weisfeiler-Lehman test.


------


## Summary of revision
**Convexity requirement:**
Following the suggestions of `7ZWa`, we have cleanly separated the conditions of activation function convexity from the proofs.
The main theorems of the paper now stand without the requirement of convex activations, albeit we note that *concave* activation functions might *break* the theorems.

**Jensen's gap:**
We have quantified Jensen's gap in Theorem 3, linking it to the variance of the effective adjacency matrix.
Particularly, the larger the variances are, the larger the gap is, which indicates slow decay of Dirichlet energy.

**Pace of presentation:**
Following the suggestions of `cENw`, we have restructured the paper so it reads more naturally and compactly now, with the problem statements interweaving with the theoretical solution.

**Discussion on limitations:**
Per the advice of Reviewer `NERp`, we have emphasized our theoretical limitations in various places, namely that we only perturb the weights of the edges without adding new edges, as that would require assumptions of the topology of the graph.

**New experiments**
We have added two new datasets to our numerical experiments, AmazonCoBuyPhoto and CoauthorCS, on both of which BRONX has achieved state-of-the-art performance, further demonstrating the utility of our model.

In sum, we have theoretically, experimentally, and presentationally improved the manuscript, based on your suggestions.
Please let us know how can further address your concerns and many thanks again for your valuable feedback.

---

### Meta-Review · Area_Chair_NDRc · 2023-12-24

**Metareview:**

This paper studies structural stochasticity and its impact on expressiveness, oversmoothing, and over-squashing. The reviewers find that the theoretical contributions are not enough. One limitation is that the quantifications of improvement are not concrete enough to illustrate the power of the proposed methods on resolving the issues of oversmoothing and oversquashing.  Another limitation is the assumption on the activation functions; though the authors argue that the convexity assumption is not necessary, it is not clear whether the analysis covers non-convex commonly used activavtion functions. Due to these limitations, I recommend reject.

**Justification For Why Not Higher Score:**

Not enough evidence to resolve the GNN issues like oversmoothing and oversquashing.

**Justification For Why Not Lower Score:**

N/A

---

### Decision · Program_Chairs · 2024-01-16

Reject